# Do Elevated Serum C-Reactive-Protein Levels Excuse Delayed Surgery for Femoral Neck Fractures?

**DOI:** 10.3390/antibiotics12040738

**Published:** 2023-04-11

**Authors:** Roberta Laggner, Benan Taner, Jennifer Straub, Thomas Manfred Tiefenböck, Harlad Binder, Thomas Sator, Stefan Hajdu, Reinhard Windhager, Christoph Böhler

**Affiliations:** Department of Orthopedics and Trauma Surgery, Medical University of Vienna, 1090 Vienna, Austria; roberta.laggner@meduniwien.ac.at (R.L.); reinhard.windhager@meduniwien.ac.at (R.W.)

**Keywords:** bone and joint infections, CRP, femoral neck fractures, hip arthroplasty, infection parameters, orthopedic infections, periprosthetic joint infection

## Abstract

In elderly patients with femoral neck fractures, preoperative serum C-reactive protein (CRP) values might be elevated due to active infections. Although there are limited data on CRP as a predictor of periprosthetic joint infection (PJI), out of concern, this could lead to delayed surgery. Therefore, we aim to investigate whether elevated serum-CRP levels justify delayed surgery for femoral neck fractures. A retrospective analysis was performed of the records of patients undergoing arthroplasty who were found to have an elevated CRP level of 5 mg/dL or more between January 2011 to December 2020. The patients were stratified to three groups, according to initial serum CRP levels at a cut off of 5 mg/dL and the time between admission and surgery (<48 vs. ≥48 h after admission). This study revealed that the patients with elevated serum CRP levels and delayed surgery showed a worse survival rate and significantly more postoperative complications than the patients on whom surgery was performed immediately. There were no significant differences in terms of PJI and prolonged wound healing in the inter-group comparison. Therefore, delays to surgery on the basis of elevated CRP values offer no benefits to patients with femoral neck fractures.

## 1. Introduction

Femoral neck fractures are a growing socioeconomic problem, and they frequently occur in elderly patients. They are associated with a high mortality rate, especially when surgery is delayed for longer than 24 h. These cases are often medically complex, and immediate surgery is not always possible [1,2,3]. For displaced fractures, hemi- or total hip arthroplasty is usually the treatment of choice [4]. For patients who live in institutional settings and/or have cognitive impairments and limited physical activities, hemiarthroplasty is the preferred option, while total hip arthroplasty is indicated in patients who have good pre-fracture levels of physical activity, self-sufficiency, and walking ability [3]. In the treatment of fractures, joint replacement is associated with higher complication rates than elective surgery [5]. Periprosthetic joint infections (PJI) are among the most feared complications in orthopedics and trauma surgery, especially in frail elderly patients, in whom they are associated with high revision and infection rates [6,7]. 

The C-reactive protein (CRP) is a cheap laboratory parameter that is widely used in clinical routines as the most important acute-phase serum protein to detect and monitor infection. It is synthesized within 6 h in response to bacterial infection, inflammatory disease, or tissue trauma in the liver and adipose tissues [8,9]. In addition, serum CRP has a long half-life and steady values with negligible circadian fluctuations [10]. In orthopedics, the relationship between preoperative serum CRP levels and the occurrence of infections in planned primary arthroplasty procedures has already been discussed. Pfitzner et al. demonstrated, in a retrospective matched-cohort analysis, that an elevated preoperative serum CRP level above 5 mg/L is an independent risk factor for periprosthetic infections in primary arthroplasty [11]. Similar studies on fracture-associated serum-CRP elevations before arthroplasty remained inconclusive [12,13]. 

In elderly patients with hip fractures, preoperative serum CRP values might be elevated due to active infections (e.g., pneumonia, urinary tract infection) or fracture-associated soft-tissue damage [5,8]. 

Although there are only limited data on serum CRP as a predictor of PJI in trauma surgery, out of concern for its complications, surgery might be delayed until serum-CRP values decrease under conservative treatment. As this approach contrasts with the usual procedure of performing surgery as soon as possible [1], a comparison of these strategies is urgently needed.

Therefore the aim of the present study was to investigate the effect of delayed hip surgery due to elevated serum CRP in relation to PJI. In addition, we also analyzed whether prolonged waiting times until surgery had an influence on other outcome parameters. 

## 2. Methods

For the design of this local-ethics-committee-approved study, we extracted patients from the database of our institution with displaced femoral neck fractures (AO/OTA 31-B) who underwent surgery and also had initially elevated serum CRP levels. A retrospective analysis was performed of the records of patients undergoing arthroplasty between January 2011 to December 2020, with a minimum follow-up of two years. Patients with other concurrent fractures, hip fractures in which the injured hip was previously treated surgically, e.g., with screws and plates, and elective admissions were excluded to avoid misclassification of the data.

Patients’ characteristics extracted from the database included: patients’ age, gender, ASA score, diabetes mellitus, anticoagulant therapy, and serum-CRP value at admission. Treatment-related information, such as time from admission to surgery, surgical technique (total hip arthroplasty or hemiarthroplasty), antibiotic treatment following surgery, and postoperative complications, including PJI, urinary-tract infection (UTI), pneumonia, 1-year mortality rate, and days of hospitalization, were extracted. 

Periprosthetic joint infection was defined according to the 2018 International Consent criteria [14]. Additionally, we recorded prolonged wound secretion and other signs of local infection within the first postoperative period. As there is no strict definition of “prolonged” or “persistent” wound drainage, persistent wound drainage was defined as wound drainage occurring or continuing 72 h after the operative procedure [15]. 

Patients were stratified into three groups according to initial serum CRP levels, at a cut off of 5 mg/dL, as well as to time between admission and surgery (< vs. ≥48 h after admission). 

C-reactive protein <5 mg/dL and surgery within 48 h (control).

C-reactive protein ≥5 mg/dL and delayed surgery delayed surgery).

C-reactive protein ≥5 md/dL and surgery within 48 h (early surgery).

At this point, patients for whom surgery was not delayed by elevated serum-CRP values but instead by other clinical reasons were excluded. Reasons for postponing surgery have to be recorded routinely due to national quality surveillance. 

The primary endpoint of the analysis was the rate of PJI. Secondary endpoints included one-year mortality rate, the overall perioperative complication rate, and the duration of hospitalization.

Blood samples for determination of serum CRP levels were obtained by peripheral venipuncture immediately before the surgical intervention. A commercially available immune-turbidimetric test (Olympus, CRP Latex, Olympus Life and Material Science Europe, Hamburg, Germany) was used for CRP-serum-level measurements. Further, an intra-assay variability between 1.64 and 3.34% is claimed by the manufacturer. Serum CRP levels of 0.5 mg/dL were defined as normal.

For statistical analysis, baseline cohort characteristics were reported as metric variables with median and interquartile range and as dichotomous variables in absolute numbers and percentages. Tests were performed using Chi-square or Fisher’s exact test for <5 observations per group. Comparisons were performed by Mann–Whitney U-test, Students’ *t*-test or Kruskal–Wallis test, as appropriate, as well as receiver-operating-characteristic curve (ROC curve) and area under the curve (AUC). All reported *p*-values were two-sided, with an alpha level of 0.05, for univariate and multivariate analyses, respectively, and were considered statistically significant. Statistical analyses were performed using the Statistical Package for the Social Sciences statistical software (SPSS 24.0 for MAC, IBMCorp., Armonk, NY, USA) and Microsoft Excel (Microsoft, Redmond, WA, USA). 

The ethics committee of Medical University Vienna approved the study protocol before data collection was started (EK no. 1999/2020). All patients gave consent to treatment according to institutional guidelines and to anonymized assessment of clinical data and treatment outcome. This was a retrospective trial. Therefore, the Institutional Review Board of the Medical University of Vienna waived the requirement to obtain distinct written informed consent from the patients.

Patient data were anonymized and de-identified prior to analysis. The study was performed according to the declaration of Helsinki, the ICH Harmonized Tripartite Guideline for Good Clinical Practice, and the guidelines of the Institutional Review Board of the Medical University of Vienna.

## 3. Results

A total of 525 patients were analyzed. The baseline characteristics of these patients are shown in Table 1. Their median age was 81 years IQR (74;87), 69.3% were female, 65.5% had an ASA Score of ≥3, 16.4% had diabetes, and 15% were on oral anticoagulation. As treatment, 421 (80.2%) patients received hemiarthroplasty and 104 (19.8%) underwent surgery for total hip arthroplasty after femoral neck fracture. 

At admission, 221 patients had serum CRP levels >5 mg/dL and 304 were selected as a control group with initial serum CRP levels <5 mg/dL and surgery within 48 h (Table 2).

According to the defined groups, the patients were stratified as follows:

C-reactive protein <5 mg/dL and surgery within 48 h (control): 304 (57.9%).

C-reactive protein >5 mg/dL and delayed surgery (delayed surgery): 109 (20.8%).

C-reactive protein >5 md/dL and surgery within 48 h (early surgery): 112 (21.31%).

Consequently, of the total 525 patients, 416 underwent surgery within 48 h, representing 78.2% of the total. 

As shown in Table 2. demographically, there were no statistically significant differences in terms of age, sex, ASA score, and implant type between the patients who had elevated serum CRP at admission and those who did not. However, there were significant differences in the rates of diabetes cases (*p* < 0.05) and in the numbers of patients on anticoagulation (*p*< 0.01) between the three groups, whereas the delayed-surgery group was characterized by a comparatively high number of patients with diabetes and anticoagulation. Significant longer hospital stays were observed in the delayed-surgery group (*p* < 0.001).

Regarding postoperative complications (Table 3), there was no significant difference in terms of PJI and prolonged wound healing in the inter-group comparison. However, there was a significant difference in the number of postoperative urinary-tract infections (*p* < 0.05) and pneumonias (*p* < 0.001). Specifically, the delayed-surgery group featured a significantly higher number of patients with urinary-tract infections compared to both the control group and the early-surgery group (control vs. delayed surgery *p* < 0.05, early surgery vs. delayed surgery *p* < 0.05). Regarding the rate of postoperative pneumonia, the delayed group showed a significantly higher number in direct comparison with the control group (*p* < 0.001).

In terms of the rates of postoperative antibiotic therapy, the numbers in the delayed-surgery group were significantly higher than those in the other groups (*p* < 0.0001). Regarding hospitalization, the group with delayed surgery stayed significantly longer in hospital (*p* < 0.001)

With regard to mortality, a significant difference was only found between the delayed-surgery group and the control group (*p* < 0.05), although there was no significant difference between the early-surgery and the control group (*p* > 0.05).

In order to determine a cut-off value for the preoperative serum-CRP levels indicating the risk of primary PJI, a ROC curve was calculated (Figure 1).

The control group with CRP <5 mg/dL and surgery within 48 h showed an AUC = 0.55. According to this curve, the serum-CRP cutoff should be set at 0.72; thus, a sensitivity of 0.88 and specificity of 0.34 were reached. For the group with serum CRP >5 md/dL and surgery within 48 h, the AUC = 0.75; therefore, the serum-CRP cutoff was set at 6.135, and a sensitivity of 0.74 and a specificity of 1.00 were achieved. Regarding the group with serum CRP >5 mg/dL and delayed surgery, the AUC = 0.55, the serum CRP cutoff was set at 9.655 mg/dL, and a sensitivity of 0.75 and a specificity of 0.55 were achieved. 

## 4. Discussion

This study revealed that patients with an elevated CRP and delayed surgery showed a significantly worse survival rate than patients with normal serum-CRP levels. Furthermore, the overall postoperative complication rate (pneumonia and UTIs) was increased in the patients with elevated serum-CRP levels and delayed surgery. However, there was no difference in the incidence of PJI between patients with elevated serum -CRP levels who had immediate surgery and those with delayed surgery. 

The mortality rate of the patients with elevated CRP and delayed surgery was significantly higher than that of the patients with normal serum-CRP levels (24% vs. 14%). In contrast, no significant survival difference was shown between the patients with elevated serum-CRP levels undergoing immediate surgery and the patients with normal serum-CRP levels. This is the first study to show a significant impact of surgery delay on survival in patients with elevated serum-CRP levels. The one-year mortality rate after femoral neck fracture ranges from20 to 40% [16,17]. This fact underlines the critical situation of patients after femoral neck fracture. This may be because patients suffering from femoral neck fracture are frequently fragile (due to old age, high rates of diabetes, or high ASA scores). The performance of surgery in fragile and geriatric patients is delicate. Additionally, these patients frequently show elevated serum-CRP levels at admission. Therefore, in daily clinical routine, many surgeons refrain from immediate surgery, perform presurgical examinations to exclude infectious disease, administer antibiotic treatment and, consequently, perform delayed surgery. This is the first study to question whether patients showing elevated CRP levels undergoing hip replacement due to fracture should undergo surgery immediately, as delayed surgery might be associated with an increased mortality rate and does not reduce the risk of PJI.

On a similar subject, previous studies revealed one-year mortality rates of 21.4% and 50% due to surgical-site infections after total joint replacement [18,19]. Therefore, PJI is one of the most feared complications of arthroplasty, leading to high morbidity and mortality [20,21,22]. In the present cohort, a total of 14 cases (2.6%) of PJI occurred. The patients with elevated CRP and delayed surgery showed a PJI in 3.6%. This rate was similarly high to that in the other two groups (2.6% serum CRP <5 mg/dL and 1.8% serum CRP >5 mg/dL and early surgery) The results of the current study contradict the data from a previously conducted trial. Buchheit et al. showed that 80% of patients with infected hemiarthroplasty had serum CRP >5 mg/dL upon admission [12]. In contrast, in our cohort, only 43% of the patients diagnosed with PJI had a serum CRP >5 mg/dL. However, this study could neither identify a cut-off for serum CRP levels to predict PJI nor establish CRP as a good preoperative predictive factor for periprosthetic joint infection. Moreover, serum-CRP measurements and consecutive examinations for inflammation were associated with delayed surgery in the study cohort and did not decrease the rate of PJI.

Zajonz et al. attempted to identify risk factors for the development of PJI in a cohort of 312 patients undergoing hemiarthroplasty after femoral neck fracture. Their study revealed preoperative serum-CRP levels, high BMI, and prolonged surgery time as independent risk factors for PJI. In addition, the study defined a cut-off for serum CRP of 1.4 mg/dL as predictive of PJI, with a sensitivity and a specificity of 69% and 70%, respectively, and a fair accuracy (AUC 0.707) [13]. While the present study did not investigate BMI and surgical time as risk factors for PJI, we could not identify a cut-off for pretherapeutic serum CRP levels to predict PJI with acceptable sensitivity. Furthermore, in the current cohort, a cut-off of 1.4 mg/dL serum CRP did not discriminate patients with PJI from patients without PJI accurately.

These diverging results could be due to the fact that in the present study, a high rate of patients with elevated serum-CRP levels and delayed surgery (70.2%) received antibiotic treatment postoperatively, while Zajonz et al. did not report the rate of patients receiving antibiotic treatment. Recent research suggests that prolonged antibiotic prophylaxis after total joint replacement, i.e., oral antibiotics for at least 7 days, results in a statistically significant and clinically meaningful reduction in 90-day infection rates in selected patients at high risk of infection [23].

As stated above, a serum-CRP level of 1.4 mg/dL was not discriminative for patients with PJI in the present study. Research suggests that CRP values between 1 and 5 mg/dL may indicate chronic inflammatory processes, malignant diseases, or rheumatoid diseases, as well as local-tissue necrosis (e.g., myocardial infarction) [24,25,26,27]. Furthermore, intracapsular femoral neck fractures are associated with a higher increase in inflammatory factors than extracapsular fractures [8]. In order to minimize the bias in the elevated serum-CRP levels caused by chronic inflammatory processes, and considering the possible fracture-associated serum-CRP-level elevation, a cut-off value of 5 mg/dL at admission was defined.

Furthermore, the diagnostic value of serum-CRP levels after primary joint arthroplasty has been discussed several times. Mederake et al. reported that serum CRP level and leukocyte count do not aid in decision making in septic two-stage-revision hip arthroplasty [28]. This statement is consistent with the research presented by Khury et al., who were also unable to demonstrate a reliable prognostic value of serum CRP levels for planning the optimal timing of replantation [29]. Moreover, Fink et al. investigated whether the serum-CRP level could be used as a screening tool to rule out late PJI; again, no predictive value was determined [30].

In addition to trauma-related tissue damage, individual patient factors may influence preoperative CRP levels. Previous studies found that female gender and older age are associated with high preoperative CRP levels, while male patients are more likely to have elevated postoperative CRP serum levels [31,32,33]. This is in line with the results of the present study, as 70% of the patients with elevated pretherapeutic serum-CRP levels were female. In addition, the patients with elevated pretherapeutic serum-CRP levels were slightly older (84 years) than the patients with normal serum-CRP levels (80 years).

Interestingly, patients with type 2 diabetes show underlying low-grade inflammation, which may be reflected by elevated serum-CRP levels [34]. In line with this hypothesis, a high rate of patients with type 2 diabetes was associated with elevated serum-CRP levels. Although diabetes is a well-known risk factor for infections [35,36], no relation to PJI or prolonged wound healing was found.

For this consecutive series of patients undergoing arthroplasty after femoral neck fracture, we found an overall rate of persistent wound drainage of 5.3%, which is in accordance with data from other trials [37]. Various studies report rates of PJI associated with persistent wound drainage ranging from 1.3% to 50%. This wide range is potentially due to the lack of a clear term or definition for persistent wound drainage and the retrospective nature of the available research [38]. Thus, as there is no consensus on the definition of persistent wound drainage, in this study, persistent wound drainage was defined as wound drainage occurring after or continuing over more than 72 h after surgery [15,37].

In general, the length of hospital stay after surgery for displaced hip fractures is associated with the number of surgeries performed by and the educational level of the surgeon [39,40]. As the current study was performed at a tertiary university hospital, many of the patients included underwent surgery by resident physicians. An analysis of the complication rates and lengths of hospital stay according to the educational level of the surgeon was beyond the scope of the current trial. The prolongation of hospital stays in the delayed-surgery patient group was most likely based on preoperative examinations to evaluate for infectious disease and antibiotic treatment.

Several potential limitations of this study have to be recognized when interpreting its results. As this was a retrospective analysis, the information gathered on antibiotic prophylaxis and inpatient complications depended on exact documentation. Although the CRP serum levels were measured in a standardized manner throughout the study, bias cannot be ruled out. Furthermore, the number of patients included in this trial was relatively small. Nevertheless, the patient records were examined rigorously, and when the data were inconclusive, the cases were excluded from the study.

## 5. Conclusions

In conclusion, we present the first study showing that delaying surgery in patients with elevated serum-CRP levels suffering from femoral neck fractures may lead to worse survival and increased complication rates. The prospective validation of the present results in future trials is warranted.

## Figures and Tables

**Figure 1 antibiotics-12-00738-f001:**
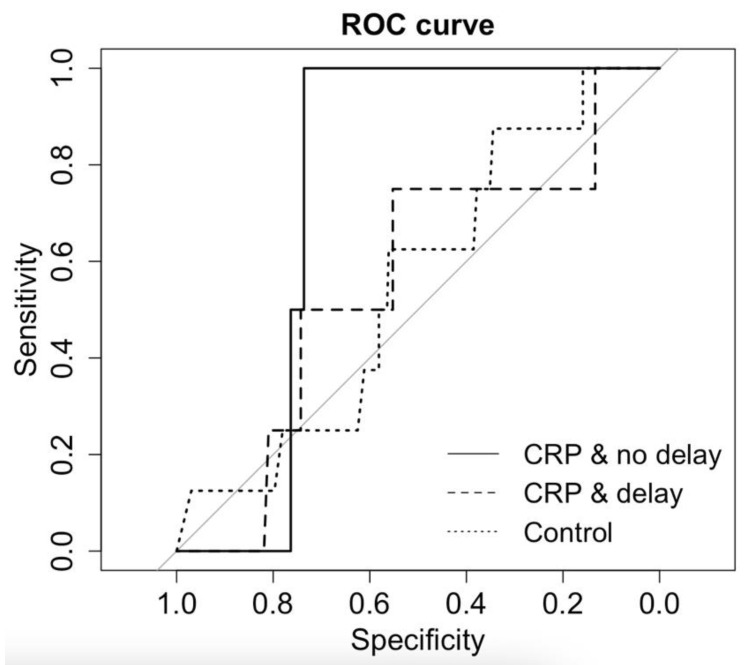
Joint confidence-region estimation for area under ROC curve and Youden index regarding serum CRP and PJI.

**Table 1 antibiotics-12-00738-t001:** Baseline characteristics.

Demographics of the Entire Patient Population (*n* = 525)	
Age (yrs) (median, IQR)	81 (74;87)
Sex, female, *n* (%)	364 (69.3%)
ASA Score *n* (%)	
ASA 1	16 (3.0%)
ASA 2	165 (31.4%)
ASA 3	315 (60.0%)
ASA 4	29 (5.5%)
Diabetes, *n* (%)	86 (16.4%)
Anticoagulation *n* (%)	70 (15%)
Implant type	
Hemiarthroplasty	421 (80.2%)
Total hip arthroplasty	104 (19.8%)

**Table 2 antibiotics-12-00738-t002:** Demographic data of the groups.

	CRP < 5 mg/dL	CRP > 5 mg/dL	CRP > 5 mg/dL	*p*-Value
	Early Surgery (*n* = 304)	Early Surgery (*n* = 112)	Delayed Surgery (*n* = 109)	
Age (yrs) (median, IQR))	80 (75;78)	84 (72;87)	83 (71;88)	0.681
Sex (female, *n* (%))	214 (70.4%)	77 (68.8%)	73 (67.0%)	0.792
Diabetes *n* (%)	45 (14.8%)	14 (12.5%)	27 (24.8%)	0.025
Anticoagulation *n* (%)	37 (12.2%)	14 (12.5%)	28 (25.7%)	0.002
ASA Score *n* (%)				
ASA 1	7 (2.3%)	3 (2.7%)	6 (5.5%)	
ASA 2	92 (30.3%)	42 (37.5%)	31 (28.4%)	0.147
ASA 3	187 (61.5%)	65 (58.0%)	63 (57.8%)	
ASA 4	18 (5.9%)	2 (1.8%)	9 (8.3%)	
CRP at admission (g/dL)	0.4 (0.1;1.12)	7.3 (6.0;9.7)	10.1 (6.7;15.8)	<0.001

**Table 3 antibiotics-12-00738-t003:** Outcome data.

	CRP < 5 mg/dL Early Surgery (*n* = 304)	CRP > 5 md/dL Early Surgery (*n* = 112)	CRP > 5 mg/dL Delayed Surgery (*n* = 109)	*p*-Value
Implant type *n* (%)				0.112
Hemiprosthesis	235 (77.3%)	92 (82.1%)	94 (86.2%)	
Totalendoprosthesis	69 (22.7%)	20 (17.9%)	15 (13.8%)	
PJI	8 (2.6%)	2 (1.8%)	4 (3.7%)	0.657
Prolonged wound healing	17 (5.6%)	4 (3.6%)	7 (6.4%)	0.667
UTI	37 (12.2%)	10 (8.9%)	22 (20.2%)	0.035
Pneumonia	5 (1.6%)	5 (4.5%)	11 (10.1%)	<0.001
Antbiotics post-OP	97 (31.9%)	32 (28.6%)	79 (72.5%)	<0.0001
Hospital stay (days, m, IQR)	16 (12;20)	16 (13;22)	23 (16;26)	<0.001
Mortality	42 (13.8%)	21 (18.8%)	26 (23.9%)	0.048

## Data Availability

Data are available from the authors on reasonable request.

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
