# Peer review of "Do Elevated Serum C-Reactive-Protein Levels Excuse Delayed Surgery for Femoral Neck Fractures?"

_antibiotics, 2023, doi:10.3390/antibiotics12040738_

Round 1

Reviewer 1 Report

The overall logic of the author's article is strong, and the conclusion is supported by solid data, but there are some flaws in the structure of the article to correct.

Author Response

Roberta Laggner

Department of Orthopedics and Trauma Surgery, Medical University of Vienna

1090 Vienna, Austria

Phone: +43 40400-59400

To:
Ms. Lyric Ren
Assistant Editor

MDPI Antibiotics Editorial Office

St. Alban-Anlage 66, 4052 Basel, Switzerland

Vienna, April 8th, 2023

Subject: Revision of: antibiotics-2333488

Thank you for the opportunity to submit our revised manuscript, entitled “Do elevated serum C-reactive protein levels excuse delayed surgery for femoral neck fractures?” (Manuscript ID: 2333488).

We have adapted the manuscript and introduced all suggested changes. Furthermore, we have corrected minor errors. Please find enclosed a detailed response to the reviews’ comments, as well as the revised version of our manuscript.

We are thankful for the valuable comments and sure that by their implementation we were able to improve the manuscript substantially. Additionally, we hope that our manuscript is now suitable for publication.

We look forward to hearing from you at your earliest convenience.

Kind regards,

Roberta Laggner and Christoph Böhler

 Manuscript: antibiotics-2333488

The manuscript is entitled “Do elevated serum C-reactive protein levels excuse delayed surgery for femoral neck fractures?”.

Review Report 1:

In this work, the authors present the study showing that delaying surgery in patients with elevated CRP serum levels suffering from femoral neck fractures may lead to worse survival and increased complication rates. Prospective validation of the present results in future trials is warranted. However, we think there are still some issues in this manuscript need to be corrected by the author before publication.

(1)The Figure 1 is too vague, please replace this image with a clear original image.

Answer: Thank you for this valuable comment: Fig. 1 was replaced with suggested by the reviewer.

(2)Sequence duplication in references, please simplify to remove.

Answer: Thank you for the observation, sequence duplication was removed.

(3)Even retrospective trials, which have investigated the use of patients' medical records, require informed consent from patients.

Answer: Thank you for your comment. In deed this was a retrospective analysis of data. The protocol of this analysis was submitted to the ethics committee of Medical University Vienna, prior to start of data acquiring. The ethics committee clearly waived the nescessity to obtain individual informed consent from patients. Furthermore, all patients treated at our clinic consent to have their data analyzed for scientific purposes.

(4)The ASA classification in Table 1 does not correspond correctly to the relevant data, so please correct it.

Answer: Thank you for the observation, the ASA score in Table 1 was corrected.

Reviewer 2 Report

This is a retrospective cohort study, conducted in a single hospital, with which the authors intend to investigate the effect of delayed hip surgery for femoral neck fractures due to elevated serum CRP in respect of PJI. Patients were stratified into three groups according to initial serum CRP levels at a cut off of 5mg/dl as well as to time between admission and femur fracture surgery. The authors conclude tant patients with an elevated CRP and delayed surgery showed a significantly worse survival than patients with normal serums CRP levels.

The work is well written, the methodology is adequate, and their experience is relevant, and it may help will help to better manage of femoral neck fractures in elderly patients and therefore I consider that the article deserves to be published with only minor changes.

However, I would suggest to the authors to consider the following minor revisions:  

1- In the statistical analysis section, on line 105 I recommend the authors delete the phrase ¨we use for statistical analysis¨ since it is repeated with what is written on line 103

2- The phrase ¨All patient records were anonymous and de-identified prior to analysis¨ is also repeated (lines 108 and 114). I recommend leaving it only in line 114

3- In the results section, lines 120-122 repeat the results of Table 1. I recommend considering not repeating. In keeping them, I think it is more correct to say that 65.5% have an ASA score equal to or greater than 3 (line 121) and the % symbol is missing after 16.4 (line 121).

4- I recommend reviewing Table 1 so that the results are aligned with the corresponding variable in the case of the ASA score

5. On line 141 Consider mentioning that the variable hospital stay appears in table 3

6. I recommend that the authors consider including a table comparing the characteristics of patients with CPR greater than and less than 5, since these differences will be discussed later in the discussion (lines 245-52).

7 Finally, I recommend reviewing the format of the bibliography since the numbering of the citations is repeated.

Author Response

Roberta Laggner

Department of Orthopedics and Trauma Surgery, Medical University of Vienna

1090 Vienna, Austria

Phone: +43 40400-59400

To:
Ms. Lyric Ren
Assistant Editor

MDPI Antibiotics Editorial Office

St. Alban-Anlage 66, 4052 Basel, Switzerland

Vienna, April 8th, 2023

Subject: Revision of: antibiotics-2333488

Thank you for the opportunity to submit our revised manuscript, entitled “Do elevated serum C-reactive protein levels excuse delayed surgery for femoral neck fractures?” (Manuscript ID: 2333488).

We have adapted the manuscript and introduced all suggested changes. Furthermore, we have corrected minor errors. Please find enclosed a detailed response to the reviews’ comments, as well as the revised version of our manuscript.

We are thankful for the valuable comments and sure that by their implementation we were able to improve the manuscript substantially. Additionally, we hope that our manuscript is now suitable for publication.

We look forward to hearing from you at your earliest convenience.

Kind regards,

Roberta Laggner and Christoph Böhler

 Manuscript: antibiotics-2333488

The manuscript is entitled “Do elevated serum C-reactive protein levels excuse delayed surgery for femoral neck fractures?”.

Review Report 2:

  • In the statistical analysis section, on line 105 I recommend the authors delete the phrase ¨we use for statistical analysis¨ since it is repeated with what is written on line 103

Answer: Thank you for the screening the phrase in line 105 was deleted.

  • The phrase ¨All patient records were anonymous and de-identified prior to analysis¨ is also repeated (lines 108 and 114). I recommend leaving it only in line 114

Answer: Thank you for the observation the phrase in line 108 was deleted, and is only mentioned in line 114.

  • In the results section, lines 120-122 repeat the results of Table 1. I recommend considering not repeating. In keeping them, I think it is more correct to say that 65.5% have an ASA score equal to or greater than 3 (line 121) and the % symbol is missing after 16.4 (line 121).

Answer: Thank you for this comment: line 120 and 121 have been corrected to 65,5% and the missing % symbol was added.

  • I recommend reviewing Table 1 so that the results are aligned with the corresponding variable in the case of the ASA score

Answer: Thank you for the note, the results in Table 1 were correctly aligned

  1. On line 141 Consider mentioning that the variable hospital stay appears in table 3

Answer: Indeed, this variable was missing, thank you for the hint, the variable “hospital stay” was added.

  1. I recommend that the authors consider including a table comparing the characteristics of patients with CPR greater than and less than 5, since these differences will be discussed later in the discussion (lines 245-52).

Answer: Thank you for this valuable comment. The characteristics of the patient groups with less than and greater than 5 mg/dl serum CRP and early vs. delayed surgery are listed in Table 2 of the manuscript. We did not include another Table comparing patient groups with less than and greater than 5 mg/dl serum CRP deliberately, to keep the manuscript concise. If required by the reviewer we can include another Table listing the characteristics solely broken down by CRP levels less or above 5mg/dl serum CRP levels as supplementary data.

7 Finally, I recommend reviewing the format of the bibliography since the numbering of the citations is repeated.

Answer: Thank you very much, we reviewed and simplified the numbering to avoid repeating citations. 
